# Low-Temperature Plasma Nitriding of Mini-/Micro-Tools and Parts by Table-Top System

**Tatsuhiko Aizawa [1],\*** , **Hiroshi Morita [2] and Kenji Wasa [3]**

1   Surface Engineering Design Laboratory, Shibaura Institute of Technology, Tokyo 144-0045, Japan
2   Nano Coat & Film Laboratory, Limited Liability Company, Tokyo 144-0045, Japan;
    hiroshi-m@mtc.biglobe.ne.jp
3   MicroTeX Labs, Limited Liability Company, Tokyo 144-0051, Japan; wasa@mui.biglobe.ne.jp
\*   Correspondence: taizawa@sic.shibaura-it.ac.jp; Tel.: +81-3-6424-8615



**Featured Application: Surface modification of mini- and micro-nozzles for dispensing systems.**

**Abstract:** Miniature products and components must be surface treated to improve their wear resistance and corrosion toughness. Among various processes, low-temperature plasma nitriding was employed to harden the outer and inner surfaces of micro-nozzles and to strengthen the micro-springs. A table-top nitriding system was developed even for simultaneous treatment of nozzles and springs. A single AISI316 micro-nozzle was nitrided at 673 K for 7.2 ks to have a surface hardness of 2000 HV0.02 and nitrogen solute content up to 10 mass%. In particular, the inner and outer surfaces of a micro-nozzle outlet were uniformly nitrided. In addition, the surface contact angle increased from 40° for bare stainless steels to 104° only by low-temperature plasma nitriding. A stack of micro-nozzles was simultaneously nitrided for mass production. Micro-springs were also nitrided to improve their stiffness for medical application.

**Keywords:** plasma nitriding; micro-nozzle; micro-spring; nitrogen supersaturation; hardening; hydrophobicity; stiffness control

## 1. Introduction

MEMS (Mechanical Electric Micro-System), miniature mechanical systems, and mini-and micro-tools have a significant risk of wear and corrosion without suitable surface treatment to each application for each component [1]. In particular, mechanical tools and parts in the order of mm and sub-mm ranges, must have sufficient wear resistance and corrosion-toughness for operations even in severe conditions [2]. Dry coatings by PVD (Physical Vapor Deposition) and CVD (Chemical Vapor Deposition) are the first policy to protect them from wearing and corrosion [3]. Their deposition layer is often limited by several to 10 μm, and inner surfaces as well as holes are difficult to coat [4]. Among some candidate alternatives, low-temperature plasma nitriding has been highlighted to provide the thick nitrided layer up to 0.1 mm with higher hardness than 1200 HV and less nitride precipitates [5–8]. This plasma nitriding at a lower temperature than 700 K was characterized by the nitrogen supersaturation; after [5], this processing was expected to be applied to various surface treatments such as carburizing and nitrocarburizing as the S-phase engineering. In addition, this nitrogen supersaturation process accompanied by two-phase nano-structuring to harden and strengthen the stainless steel parts and members as pointed in [8,9]. As demonstrated in [9–11], the corrosion toughness was also improved in these nitrogen supersaturated stainless steels. Furthermore, inner surfaces and small holes in the dies and punches are efficiently nitrided and hardened with more ease than coatings [12,13]. These intrinsic features of low-temperature plasma

nitriding were accommodated to miniature tools, and even parts, by scaling down the chamber size [14,15]; e.g., the hollow cathode device assisted the high-density plasma nitriding process in the smaller chamber systems.

In the present paper, a table-top plasma nitriding system is developed for low-temperature plasma processing of mini- and micro-tools. Micro-nozzles as well as micro-springs are plasma nitrided at 673 K for 7.2 ks to describe their nitriding and hardening behavior. In particular, a AISI316 micro-nozzle specimen is utilized to analyze the nitrogen super-saturation on the outer and inner surfaces as well as its outlet. In addition, its surface contact angle is measured to prove that the nitrided surfaces become hydrophobic. Furthermore, a stack of micro-nozzles is also homogeneously nitrided to demonstrate the capacity of the present system in application to industries. Two micro-springs are also nitrided to demonstrate that they have higher stiffness by 10 % than before nitriding. The present table-top system works to locally increase the stiffness and hardness of various tools and parts by selective plasma nitriding.

## 2. Experimental Procedure

### 2.1. Down-Sized Plasma Nitriding System

The high-density RF (Radio-Frequency)/DC (Direct Current) plasma nitriding system is downsized by 1/20 as suitable equipment for surface treatment of miniature mechanical elements in the dimensional size range from mm to sub-mm. Figure 1a depicts a table-top plasma nitriding system with a chamber size of φ350 mm × 150 mm. This chamber is automatically operated to open and move upward for the experiment setup. Every experimental operation is performed on the touch-panel in the controller. All commands as well as data acquisition are driven by a process computer through wireless communication.

There are two setting-up modes in this system, for nitriding a single component and for simultaneously nitriding a stack of elements up to 24 pieces. In the following, the former setup is used to describe the nitriding behavior of a single micro-nozzle. The latter is also employed to demonstrate the feasibility of simultaneous nitriding in mass production.

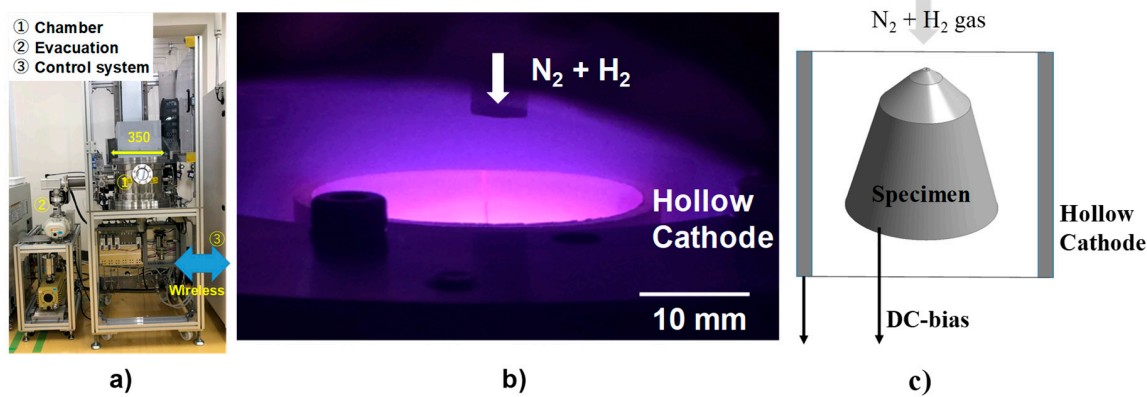

**Figure 1.** Down-sized plasma nitriding for surface treatment of the miniature mechanical elements. (**a**) Experimental apparatus, (**b**) hollow cathode to intensify the ion density in $N_2 + H_2$ plasmas, and (**c**) illustration on the hollow cathode device.

The hollow cathode device was utilized to increase the $N_2^+$ ion and NH-radical densities in the experimental setup to work in both modes. Figure 1b depicts an experimental setup for nitriding of a single micro-nozzle. As also illustrated in Figure 1c, $N_2 + H_2$ mixture gas was introduced to this DC-biased hollow. Since the DC-bias was also applied to this micro-nozzle, the ignited RF (Radio Frequency)-plasma was also confined inside the micro-nozzle by the hollow cathode effect.

## 2.2. Nitriding Conditions

The plasma nitriding conditions are summarized in Table 1. The DC-bias is parametrically varied to investigate the sputtering effect on the nitriding process. After [9], the nitrogen and hydrogen flow rate ratio was controlled to be constant by 160 mL/min for nitrogen and 30 mL/min for hydrogen, respectively. In situ plasma diagnosis in [9] revealed that the highest NH radical density against $N_2^+$ ion density was yielded around this flow rate ratio.

**Table 1.** High-density plasma nitriding conditions.

| Item | Parameters |
|---|---|
| RF-Voltage | 250 V |
| DC-bias | −300 V, −400 V, −500 V |
| Pressure | 70 Pa |
| Temperature | 673 K |
| Duration | 7.2 ks |

## 2.3. Specimens

Austenitic stainless steel type AISI316 plates and micro-nozzles were prepared for this low-temperature plasma nitriding. Figure 2a depicts a typical micro nozzle for a dispensing system with the outlet diameter of 1 mm at the top. AISI316 plate with a diameter of 25 mm and thickness of 5 mm, was employed to describe the nitriding and hardening behavior by using this table-top plasma nitriding system. Two types of micro-springs were also prepared for nitriding to improve their stiffness, as shown in Figure 2b.

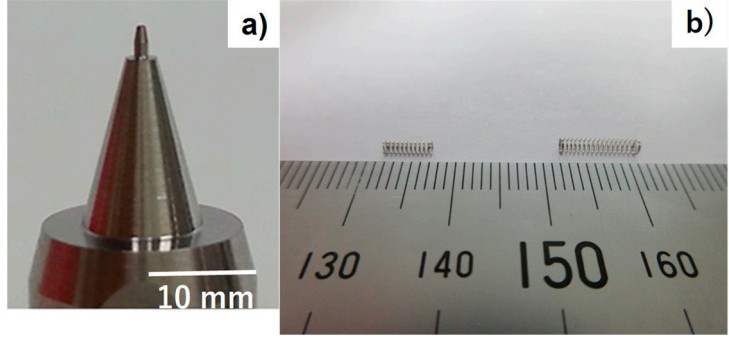

**Figure 2.** Miniature, mechanical parts for plasma nitriding. (**a**) Micro-nozzle with the outlet diameter of 1 mm, and (**b**) micro-springs for medical applications.

## 2.4. Observation and Measurement

SEM (Scanning Electron Microscopy; JSDM-IT300LV, JEOL Ltd., Akishima, Tokyo, Japan) with EDX (Electron Dispersive X-Ray Spectroscopy; Pegasus, EDAX, Inc., Minato-ku, Tokyo, Japan) were utilized to describe the nitrided specimen and to analyze the nitrogen solute content distribution on the surfaces. Micro-Vickers hardness testing (HM-210C; Mitsutoyo Co., Ltd., Kawasaki, Japan) was also employed to measure the surface hardness for various applied weights.

## 3. Results

### 3.1. Nitriding Behavior of AISI316 Substrates

AISI316 plate was nitrided at 673 K for 7.2 ks by parametrically varying the DC-bias as listed in Table 1. The nitrogen ion density as well as the NH radical density increases with increasing the DC-bias. More nitrogen atoms penetrate from the substrate surface and diffuse into its depth [6,9]. Figure 3a depicts the variation of measured nitrogen solute content by EDX with increasing the DC-bias.

The nitrogen solute content at the surface increases monotonically with DC-bias. In the following nitriding experiments, this DC-bias is fixed to be constant by −500 V. The surface roughing took place when this DC bias became lower than −500 V.

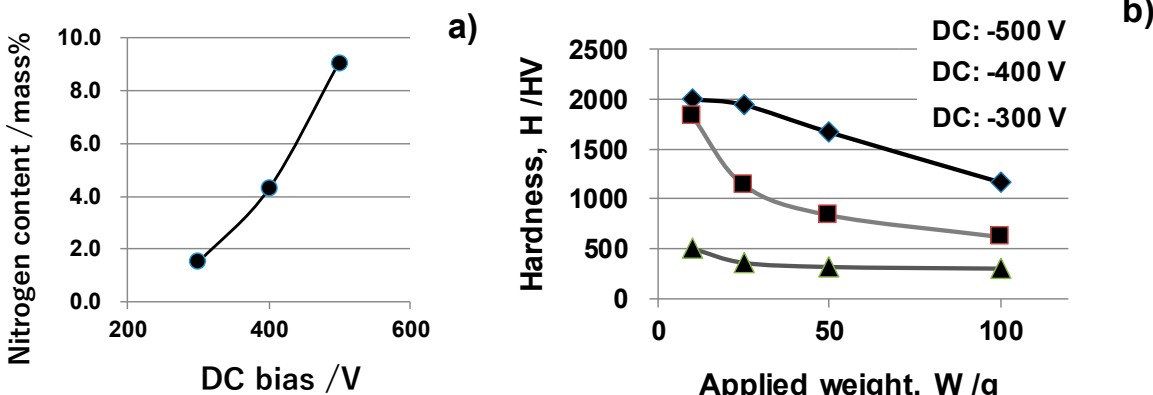

**Figure 3.** Effect of DC-bias on the nitrogen content and hardness. (**a**) Variation of the nitrogen solute content, and, (**b**) variation of hardness profile.

In the micro-Vickers hardness testing, the matrix hardness affects the measured hardness when the indentation depth exceeds (the nitrided layer thickness)/6 [12,16]. In fact, the measured hardness of thin nitrided layer became lower than the surface hardness by increasing the applied weight in testing [12,13]. Figure 3b depicts the variation of hardness with the increasing applied weight in hardness testing. In case of the nitriding by DC-bias of −400 V, the hardness significantly reduces toward the matrix hardness of 250 HV with an increase in the loading weights. Higher DC-bias is also needed to attain a nitrided layer sufficiently thick enough to keep high hardness in depth.

### 3.2. Nitriding of a Single Micro-Nozzle

AISI316 micro-nozzle specimen in Figure 2a was also plasma nitrided at 673 K for 7.2 ks. Figure 4a shows the nitrided micro-nozzle. The spot for EDX was controlled to move from the edge of the outlet toward the step of the nozzle body along the line-A in Figure 4a. Figure 4b depicts the nitrogen solute distribution along this line-A, from the top of the outlet down to the nozzle body. Both the nozzle outlet and the nozzle body surfaces are uniformly nitrided to have high nitrogen contents, except for the vicinity of a step between two regions. On the surface of this step, the electron beam in EDX significantly scattered to lower the measured intensity. This average intensity in Figure 4b corresponds to the nitrogen content by 8.6 mass%. This nitrogen content is equivalent to the measured content of 9 mass% for the nitrided AISI316 plate in Figure 3a.

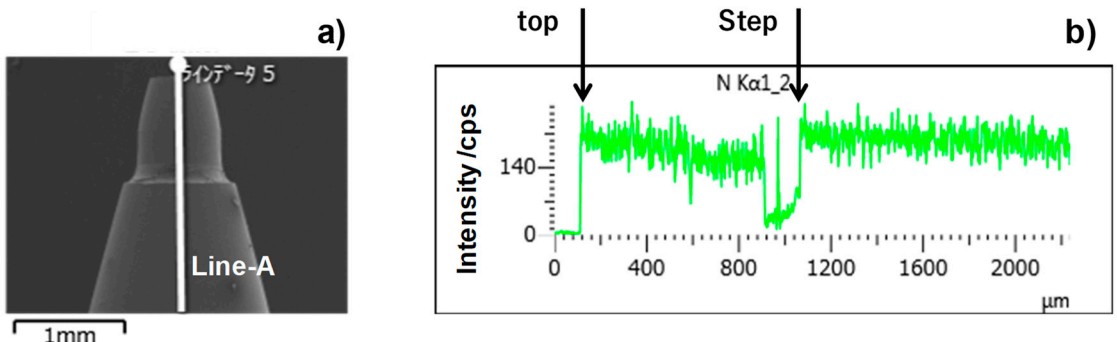

**Figure 4.** Plasma nitrided AISI316 micro-nozzle at 673 K for 7.2 ks. (**a**) Outlook of nitrided micro-nozzle, and, (**b**) nitrogen mapping measured from the top of the outlet down to the nozzle body.

As had been discussed in [12,13], the inner surface of the outlet channel in the micro-nozzle could be hardened when using the nozzle channel as a hollow cathode. A single micro-channel with the outlet diameter of 0.2 mm was also used as the hollow cathode in the present plasma nitriding to evaluate the hardness across the nozzle thickness.

Figure 5b depicts the hardness distribution across the thickness of nitrided micro-nozzle. The average hardness becomes more than 1000 HV0.1 at the vicinity of the channel toward the outlet. The surface hardness of the nitrided AISI316 plate by 100 g or 1 N, becomes 1000 to 1100 HV in Figure 3b. This suggests that a single micro-nozzle is nitrided and hardened to be 1000 HV not only on its outer surface but also on its inner surface even at 673 K for 7.2 ks.

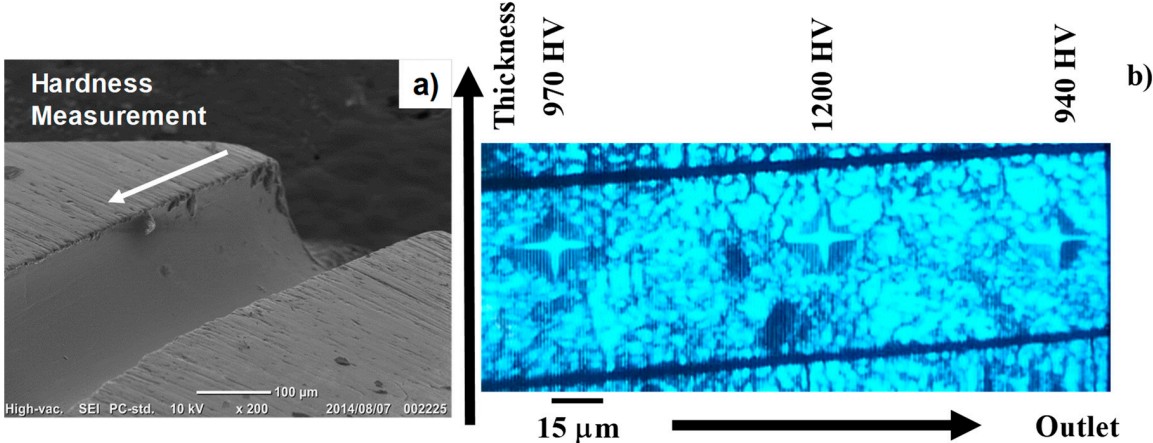

**Figure 5.** Inner nitriding of the nozzle hole. (**a**) A cross-section of plasma nitrided micro-nozzle with the outlet diameter of 0.2 mm, and, (**b**) micro-hardness distribution near the inner surface of nitrided micro-nozzle at 673 K for 7.2 ks.

### 3.3. Wettability Control by Plasma Nitriding

Bare stainless steel is hydrophilic as its contact angle against the pure water is only 40°. This is because the metallic surface including the stainless steels has higher surface energy. The contact angle measurement was performed to modify this surface property by nitriding the micro-nozzle for dispensing the droplet from the outlet hole with low adhesion. Figure 6 depicts the pure water, swelling on the nitrided AISI316 plate at 673 K for 7.2 ks. The original hydrophilic surface changes itself to a hydrophobic surface with the contact angle (θ) of 104°. This suggests that this micro-nozzle outlet could be hydrophobic enough to dispense the droplet with low adhesion to the outlet [12,15].

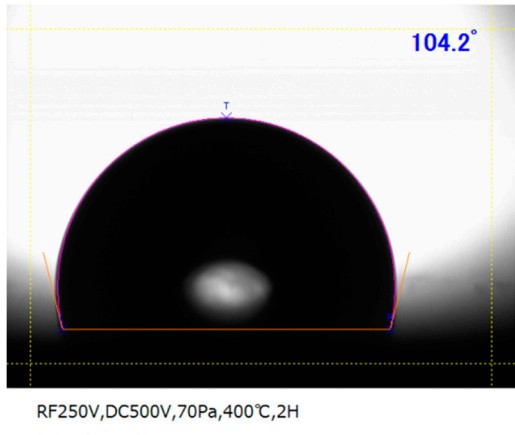

**Figure 6.** Measurement of the contact angle for a pure water droplet on the nitrided AISI316 surface.at 673 K for 7.2 ks.

### 3.4. Simultaneous Nitriding of Micro-Nozzles

This table-top plasma nitriding system can simultaneously nitride multiple micro-nozzles in a stack. Using the rotating mechanism in Figure 7a, twenty-four micro-nozzles were nitrided at 673 K for 7.2 ks at the same plasma sheath conditions. EDX was utilized to measure the nitrogen content for each micro-nozzle in a single stack.

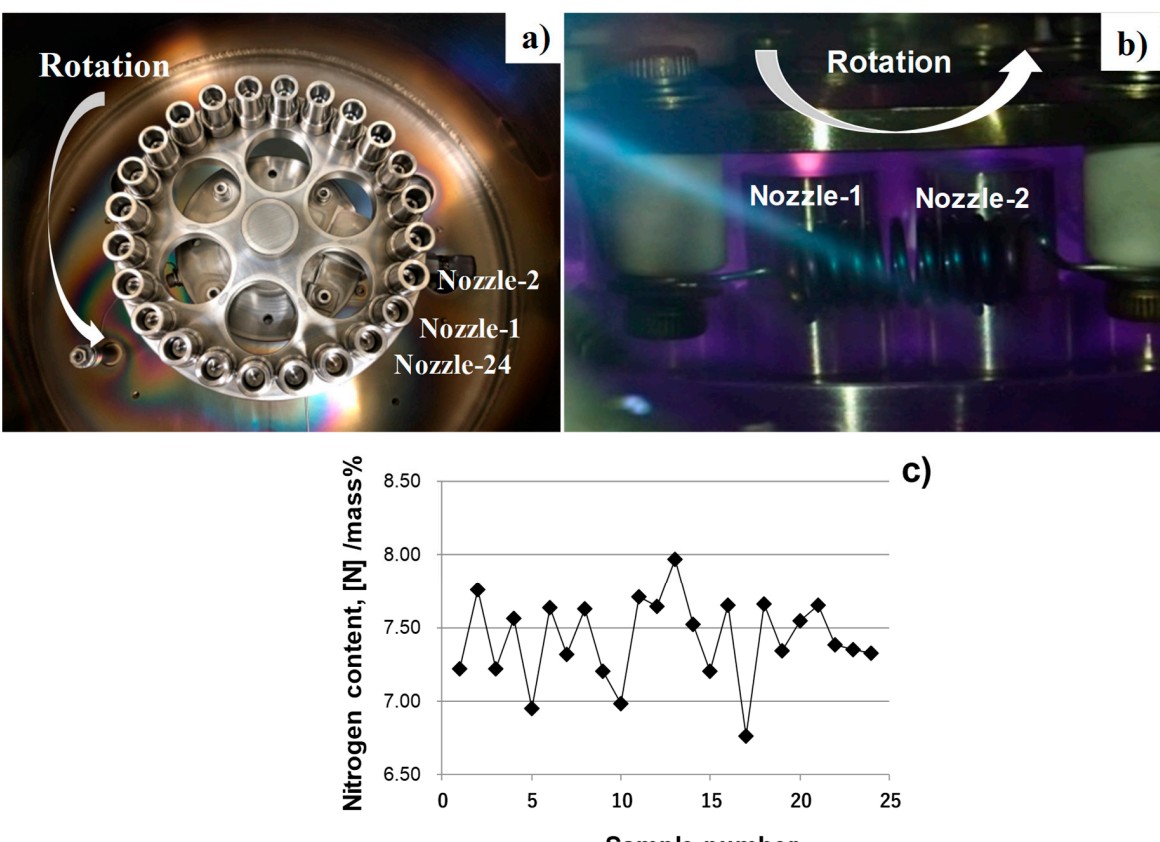

**Figure 7.** Simultaneous nitriding behaver of 24 micro-nozzles. (**a**) Experimental setup of 24 micro-nozzles into the DC-biased plate with heaters, (**b**) An experimental set-up for simultaneous plasma-nitriding of 24 micro-nozzles, and, (**c**) Statistical distribution of the surface nitrogen content among 24 nitrided micro-nozzles at 673 K for 7.2 ks.

Three positions were selected for this EDX analysis on each micro-nozzle surface; e.g., the top of the outlet, the step between the outlet and body, and, the center of the body in Figure 2a. Figure 7a shows the initial setup of 24 micro-nozzles into the DC-biased plate with electric hears. As depicted in Figure 7b, the stack of micro-nozzles is rotated in the plasma sheath. Figure 7c shows the measured nitrogen content for each micro-nozzle by EDX. A deviation of average nitrogen contents is low among twenty-four micro-nozzles in the same stack. After statistical analysis, the mean nitrogen content is estimated to be 7.4 mass% and the standard deviation, 0.3 mass%. Nitriding took place homogeneously and is the equivalent to the nitriding of a single micro-nozzle in Figure 3.

In addition to this nitrogen mapping, the chromium content was also measured; the average content is 16.4 mass% and its standard deviation, 0.4 mass%. This proves that chromium content in these nitrided micro-nozzles remains the same as before nitriding and that no CrN is precipitated by this plasma nitriding.

### 3.5. Simultaneous Nitriding of Micro-Springs

The micro-spring constant (k) is determined by its geometry and number of turns in its length. Since its diameter, thickness, and length are strictly specified in each medical application, k is difficult to control once it is fabricated as a spring. Two micro-springs were prepared to demonstrate the possibility to increase their spring constants by nitriding their inner surfaces; e.g., a shorter microspring-1 and a longer micro-spring-2 as listed in Table 2.

**Table 2.** Geometry and dimensions of two micro-springs.

| Item | Microspring-1 | Microspring-2 |
|---|---|---|
| Wire diameter (d) | 0.2 | 0.15 |
| Coil diameter (D) | 1.7 | 1.5 |
| Length (L) | 5.7 | 9.5 |
| Number of turns (N) | 10 | 16 |

One end of micro-spring was fixed to the metallic jig, to which DC-bias was applied. Its inner cylindrical space was utilized as a hollow cathode to nitride its inner surfaces at 673 K. The duration time was also constant by 7.2 ks. It is difficult to measure the hardness and microstructure of thin wire in these micro-springs. Their spring constant is employed to evaluate the strengthening by plasma nitriding of the spring wires. The uniaxial tensile testing with small-scaled load cell was employed to measure the applied load with the minimum resolution of $10^{-3}$ N or 0.1 gram-weight. Figure 8 depicts the load-displacement relations between an original spring and two nitrided micro-springs in type-1. The spring constant is 9.3 mN/mm for the original microspring-1, while k = 10 mN/mm for the nitrided one. The spring constant is enhanced by 8% through the plasma nitriding. In case of the microspring-2, k = 7.3 mN/mm is increased up to 8.0 mN/mm by nitriding. This increase of spring constant by 8 to 10 % reveals that the inner surface nitriding has a significant influence on the strengthening of micro-springs.

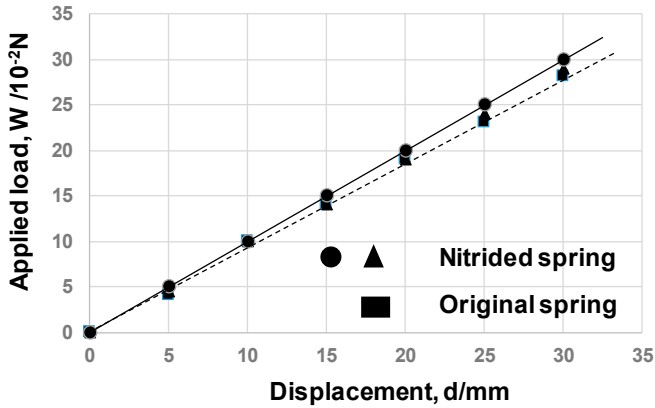

**Figure 8.** The load–displacement relationship for the original spring and two nitrided springs.

## 4. Discussion

In the deposition process by PVD (Physical Vapor Deposition) and CVD (Chemical Vapor Deposition) coating onto the inner surfaces, the adhesion probability of coating particles is high enough to coat at the vicinity of micro-nozzle inlet but significantly decreases with the distance toward the outlet [17]. Hence, it is very difficult to deposit the inner surfaces of a micro-nozzle hole smaller than 1 mm in diameter and to coat the curved surfaces in the micro-spring with uniform thickness. In the present plasma nitriding, the inner surfaces of the micro-nozzle as well as micro-springs are subjected to the plasma sheath by the hollow cathode effect. Small RF-plasma is confined to the inside of the micro-nozzle by the application of DC-bias. As shown in Figure 5, its inner surface is nitrided and

hardened up to 1000 HV. This reveals that the tool life of micro-nozzles could be prolonged by mini- and micro-sized plasma control for nitriding of their inner surfaces [12,13,18].

The stiffness of the coil-springs (k) is defined by the following equation,

$$k = G \times d^4/(8 \times N \times D^3), \tag{1}$$

where G is the effective shear modulus of the stainless steel wire including the residual stresses in the fabrication of springs from wires and so forth. Since the micro-springs were self-standing on the DC-biased metallic fixture during nitriding, the geometry of coil spring and number of turns are insensitive to the perturbed modification by nitriding; e.g., $\Delta D = \Delta N = 0$ in nitriding. The ratio of perturbed stiffness ($\Delta k$) to k is then given by a variation of Equation (1),

$$\Delta k/k = \Delta G/G + 4 \, (\Delta d/d), \tag{2}$$

where $\Delta G$ is a perturbed effective shear modulus and $\Delta d$, a perturbed wire diameter.

Since $\Delta d$ is negligibly small in measurement, the main contribution to the increase of ($\Delta k/k$) in Equation (2) is ($\Delta G/G$) during nitriding. Consider that the torsional shear strain is induced on the cross-section of the stainless steel wire in the coil spring during its elongation or compression. The inner surface of the micro-coil is hardened by nitriding to reduce this torsional strain even when applying the same load to the micro-springs. This reveals that the equivalent shear modulus of wires is significantly increased up to 8 to 10 % by the inner nitriding to enhance the stiffness of micro-springs. In medical applications, a top-edge of tweezers and forceps as well as these micro-springs might well have higher stiffness to catch and hold the targeted objects precisely in operation. Selective nitriding of these tools by the present system provides a means to control their stiffness in part.

## 5. Conclusions

A table-top high-density plasma nitriding system is developed to super-saturate a single AISI316 micro-nozzle, as well as a stack of AISI316 micro-nozzles, with nitrogen. DC-bias is optimized to be −500 V to fabricate the nitrided micro-nozzles with a higher surface nitrogen content than 8 mass% and higher hardness than 1200 HV0.1 by nitriding at 673 K for 7.2 ks. In particular, the hardness at the depth of 40 μm reaches to 1000 HV0.1. Due to this nitrogen super-saturation into the depth, the original hydrophilic nozzle surface can be modified to be hydrophobic with $\theta > 100°$. This high content nitrogen super-saturation also takes place homogeneously even for a stack of 24 micro-nozzles. The present table-top system is also useful to increase the stiffness of micro-springs by 10%. Selective hardening of inner surfaces results in the significant increase of equivalent shear modulus in wires of springs. This proves that the present system works to control the local stiffness in various medical tools and parts by selective nitriding.

**Author Contributions:** Design, conceptualization and methodology of miniature nitriding system by T.A.; experiments by K.W. and H.M.; writing-original draft preparation by T.A.; writing—review and editing by this T.A. and H.M.

**Funding:** This study was financially supported in part by MEXT-project on the supporting industries in 2018.

**Acknowledgments:** The authors would like to express their gratitude to S. Hashimoto (TECDIA, Co. Ltd., Japan), A. Farghali and S. Kurozumi (SIT) for their help in experiments.

**Conflicts of Interest:** The authors declare no conflict of interest.

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
