# Peer review of "Low-Temperature Plasma Nitriding of Mini-/Micro-Tools and Parts by Table-Top System"

_applsci, doi:10.3390/app9081667_

Round 1
Reviewer 1 Report
The concept of compact-size nitriding system will be of industrial importance so that the referee concluded that the relevant development deserves to be published after minor revision.
We hope that the following comments are of help to brush up the paper.
A detailed illustration about the electrode, hollow cathode, etc. is helpful to understand the device.
The fraction of N2 and H2 and the optimization of their ratio should be mentioned.
Please add some references of previous works on the effect of stainless-steel nitriding on hydrophilicity. If your result is the first, please specify the novelty.
Author Response
THe manuscript was revised with consideration on the request by the reviewer-1.

Reviewer 2 Report
This paper reported a well-developed technology of low temperature plasma nitriding of austenitic stainless steels on MEMs application with authors’ own hollow cathode settings. Two successful case studies are of interesting and could attract industry to mimic the treatment conditions for similar MEMs components.
However, the paper is needed to be further improved for publishing in the Journal concerned. Hereby detailed comments are given for the benefit of the authors in modifying this paper.
1. Introduction part needs to be expanded in terms of the importance of the ‘S-phase’ researches and the variety of the researches, such as nitriding, carburising and nitricarburising.
2. The novelty of current research also needs to be further emphasized.
3. Experimental: a schematic hollow cathode setup for nitriding of a single micro-nozzle is needed to explain the Fig 1b.
4. The equipment used for the microstructure observation, EDX analysis and hardness measurements need to be specified.
5. How temperatures were measured for the hollow cathode treatment of the nozzle hole? Any good SEM images of the surface treated layer can be provided as the Fig 5 b shows very poor metallography of the surface layer?
6. A schematic of simultaneous nitriding of 24 micro-nozzles is also needed for understanding Fig 7 a, as the picture does not give clear clue about the set-up.
7. Again, as for the micro-springs, a metallurgical image of the treated surface layer would give a strong evidence of the successful treatment.
8. Above number 5 and 6, or SEM images of the treated surface layer are very important to check if any CrN precipitates formed during the hollow cathode plasma nitriding treatments. This is one of the very important aspects of the treatment. Should the CrN precipitated, the corrosion resistance of the stainless steel will be degraded and the application of the treated component will be limited or even prohibited
Author Response
The original manuscript was revised with nearly full consideration on the request by the reviewer-2.
